**euroCRIS Track at the 2021 VIVO annual conference: "VIVO implementations at Latin American institutions: different approaches and configurations, one common objective"**

The euroCRIS Directory of Research Information Systems (DRIS) currently displays a good number of VIVO implementations in many different countries. In previous euroCRIS Tracks at VIVO annual conferences the quick rate of growth for VIVO implementations in Latin America was specifically highlighted, together with the quality of some of these. This was also pointed out during a recent Latin American panel discussion on the emergence of research information management systems in the region. The strong reliance of these developments on open source software solutions could also significantly enhance the opportunities for cross-institutional and international collaboration. After having highlighted a number of CRIS-related VIVO implementations in Europe during previous euroCRIS Tracks at the 2019 and 2020 VIVO annual conferences, this year we would like to focus on a couple of these VIVO implementations in Latin America.

These case studies will be complemented with an introductory presentation on the growing presence of Latin American CRIS systems in the euroCRIS Directory of Research Information Systems (DRIS) and with a brief interview with the lead for the recently formed Spanish-language VIVO User Group.

As for the system configuration for the Latin American VIVO instances featured in the session, case studies will be shown for VIVO used as *the* institutional CRIS system and for VIVO operated as a public research portal on top of a 'monolithic' CRIS system – a diversity of approaches similar to the European examples that were examined last year. The session will explore these cases in more detail via presentations from two Latin American institutions in Mexico and Colombia. The session will finish with a round of Q&A during which the specifics of CRIS development in Latin America will be addressed.

The planned structure for the session is as follows:

• "Latin American CRIS systems in the euroCRIS Directory of Research Information Systems (DRIS) with an emphasis on open source software solutions" (Pablo de Castro, euroCRIS Secretary)

• 15-min interview with Anna Guillaumet (euroCRIS Board member and VIVO Leadership Group), Lead for the Spanish-language VIVO User Group. Interviewer: Pablo de Castro

  Latin American VIVO implementations: case studies

• "ORBIS: VIVO as an institutional CRIS Portal at the Autonomous University of San Luis Potosi, Mexico" (Irene Carmen Portillo Vázquez, Claudia Lizett Amaya Varela, Universidad Autónoma San Luis Potosí, México)

• "HUB-UR Services and Experts Finder: a VIVO-based System on top of the institutional CRIS at Universidad del Rosario (Colombia)" (Maria Lucia Lizarazo, Universidad del Rosario, Colombia)

• Round of Q&A on the specifics of CRIS development in Latin America