# OpenReview forum: "euroCRIS Track at the 2021 VIVO annual conference: “VIVO implementations at Latin American institutions: different approaches and configurations, one common objective”"
_vivoconference.org/VIVO/2021/Conference_

### Official Review · Program_Chairs · 2021-05-29
**Critically important examples to share and discussions to be had**

**Rating:** 10
**Confidence:** 5

**Review:**

The authors propose a track to consider important examples of VIVO implementations in Latin America and consideration for future growth of VIVO in the region.  The authors are experienced with the issues and with organizing such conversations.

This critically important track should be a key contribution to the conference in 2021.

Some questions that might be important for the participants to consider: 1) how can developers from Latin American become involved in the improvement of the software for use across the region?; 2) how can leaders from Latin America become leaders in the VIVO community; 3) how can Latin American community needs be assessed so that these needs might be addressed in future versions of VIVO?

This is an outstanding track proposal that should serve as a catalyst for the improvement of VIVO in service to the Latin American community.